# Real-Time In-Situ Investigation of the Neutron Irradiation Resistance Ability of Nd$^{3+}$-Doped Gd$_3$Sc$_2$Al$_3$O$_{12}$ Laser Crystal

Yuxi Gao [1,2], Wenpeng Liu [1,*], Shoujun Ding [1,3], Yuanzhi Chen [4] and Qingli Zhang [1,*]

1 Anhui Institute of Optics and Fine Mechanics, Chinese Academy of Sciences, Hefei 230031, China
2 University of Science and Technology of China, Hefei 230026, China
3 School of Science and Engineering of Mathematics and Physics, Anhui University of Technology, Maanshan 243002, China
4 School of Material Science and Engineering, Shanghai Institute of Technology, Shanghai 201418, China
* Correspondence: wpliu@aiofm.ac.cn (W.L.); zql@aiofm.ac.cn (Q.Z.)

**Abstract:** In optical crystals, photodarkening will occur after they were irradiated with high-energy particles, and such induced optical loss generally results in significant performance degradation whether they are used as passive or active optical elements. In the present study, the effects of neutron irradiation on the optical response of the Nd$^{3+}$-doped Gd$_3$Sc$_2$Al$_3$O$_{12}$ (Nd:GSAG) single crystal has been revealed in real-time and in-situ. Transient and permanent transmittance reduction in the crystal induced by neutron radiation has been observed and the reduction mechanisms have been analyzed. The XRD characterization method demonstrated that the crystal structure remained constant both before and after neutron irradiation. Importantly, the X-ray photoelectron peak of the O 1s core level shifts to high binding energy, indicating that oxygen vacancies were produced in the crystal after irradiation with neutrons. Thus, the permanent reduction in the transmittance of the crystal after irradiation with neutrons can be attributed to the generation of oxygen vacancies in the crystal. To the best of our knowledge, it is the first time the damage types in rare earth oxide laser crystals caused by neutron irradiation were revealed.

**Keywords:** Nd:GSAG crystal; in-situ and real-time; neutron irradiation; X-ray photoelectron spectra; oxygen vacancy

## 1. Introduction

Nowadays, with the advances of space technology, space-based laser systems have been extensively established for applications such as environmental detection, space-ground communication, remote sensing technology, etc. [1] The LD-pumped all solid-state laser (ASSL) system, owing to its simplicity, compactness and durability, makes it more suitable for deployment and application in the space environment. As we know, outer space is distributed with space radiation induced by high-energy particles such as electrons, γ-rays, protons and neutrons, and thus the laser crystal working in the ASSL system will encounter both high-flux transient and low-flux continuous-wave space radiation after being launched into outer space [1–3]. The laser crystal irradiated with space radiation can result in the production of ionization damage, displacement damage, and structural damage inside the crystal [4,5]. Therefore, one of the challenges inherent in the deployment of ASSL systems in space is the potential for radiation-induced damages and changes to optical properties of the laser crystal during its exposure to a high-energy radiation environment. Although shielding can reduce the effects of exposing the laser crystal to low-energy particle radiation, it is quite difficult to protect the crystal against exposure to high-energy radiation. Thus, the investigation of the effect of high-energy radiation on the optical transmittance of the laser crystal is of great interest for its application in adverse radiation environments, which has attracted much attention.

The effects of irradiation on the optical properties of crystals have been reported elsewhere. For example, Rai et al., irradiated the Nd-doped phosphate glass with 5–500 kGy doses of gamma rays and found that the transmittance of Nd-doped phosphate glass was reduced significantly even irradiated with 5 kGy doses of gamma rays [6]. Sun et al., revealed the effect of 100 Mrad gamma-ray irradiation on the transmittance and luminescence properties of Cr:GSGG crystal [7]. In our previous work, we also reported the gamma-ray irradiation on the transmittance and luminescence properties of Nd:GSAG and Nd:YSAG crystals [2,8,9]. However, the primary mechanism for the radiation-induced changes in the optical properties is not fully understood, and controversies still exist currently.

From the results obtained by electron paramagnetic (EPR) and thermoluminescence techniques, there are at least three types of trapping centers induced in optical crystals (yttrium aluminium garnet) by gamma radiation: (I) electrons caused by the six non-equivalent orientations of the oxygen tetrahedrons in the garnet lattice and a weak interaction of an unpaired electron with the nuclei; (II) $O^-$ type centers coordinate the $Al^{3+}$ ions in a-sites; (III) oxygen vacancies formed during the crystal growth, while one of them was found to be stable at room temperature [1,10]. This means, if the optical loss induced by the radiation cannot be investigated in real-time, then the research conclusion may not be accurate. Hence, a setup that can observe the crystal transmittance and varies the neutron radiation dose in real-time was built in this work. Using this setup, the effect of neutron radiation on the transmittance of Nd:GSAG crystal was investigated in real-time for the first time, to the best of our knowledge. Transient and permanent transmittance reduction of the crystal induced by neutron radiation has been observed. Additionally, the crystal structure and the valence state of the elements in Nd:GSAG were also investigated after irradiation with neutron radiation.

## 2. Experimental Section

### 2.1. Crystal Growth

High-quality 1 at% $Nd^{3+}$-doped GSAG crystal was grown in a JGD-80 furnace (26th institute of CETC) by the Cz method. Powder compounds $Nd_2O_3$ (5N), $Gd_2O_3$ (5N), $Sc_2O_3$ (5N), and $Al_2O_3$ (5N) were employed as the raw materials. The oxides were weighed according to the chemical formula $Nd_{0.03}Gd_{2.97}Sc_2Al_3O_{12}$. When the crystal growth procedure was completed, the crystal was cooled down to room temperature at a rate of about 30 K/h. Finally, the crystal was cut into pieces of different sizes with a good parallel and polished on both sides for the following experimental study.

### 2.2. In-Situ Setup

The neutron radiation appliance was a High-Intensity D-T Fusion Neutron Generator, which was designed and established by the Fusion Design and Study team from the Institute of Nuclear Energy Safety Technology, Chinese Academy of Sciences. The setup mainly includes a high-intensity ion source, an accelerating tube, and a high-power rotating tritium target. The mechanism of production and the property of the fusion neutron was demonstrated previously by Wu [11]. An in-situ real-time transmission measurement device was built at the outlet of the neutron source. The schematic diagram and physical map of this system for in-situ and real-time investigation of the neutron irradiation resistance ability of Nd:GSAG single crystal are shown in Figure 1a,b. A steady radiation mode was employed for the generation of fusion neutrons with a neutron yield of $1.1 \times 10^{11}$ n/s. The Nd:GSAG crystal was placed about 7 cm away from the neutron outlet, corresponding to a neutron radiation rate of 7.3 mGy/s.

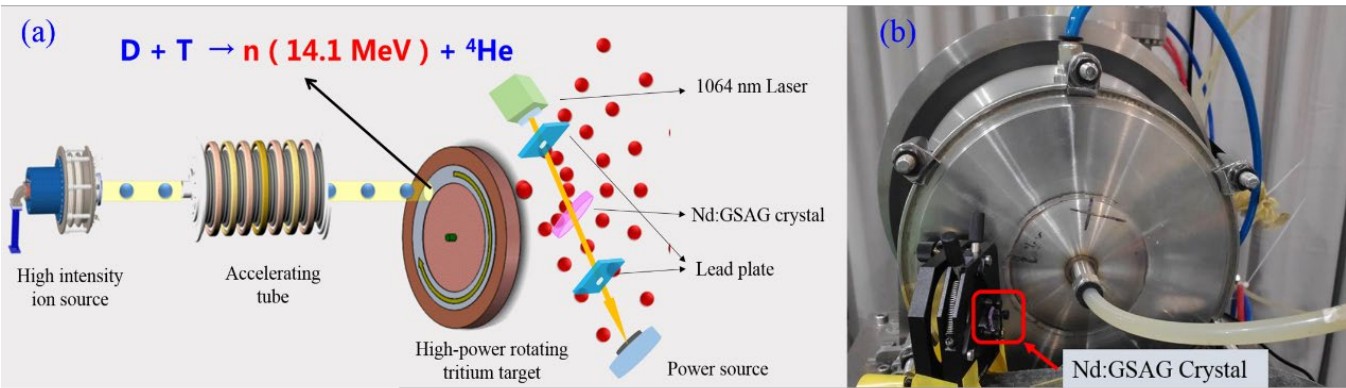

**Figure 1.** (**a**) The schematic diagram of the system for in-situ and real-time investigation of the neutron irradiation resistance ability of Nd:GSAG single crystal. (**b**) The physical map of the system for in-situ and real-time investigation of the neutron irradiation resistance ability of Nd:GSAG single crystal.

*2.3. Characterization*

The absorption spectrum was measured at room temperature with a PerkinElmer UV-VIS-NIR lambda-950 spectrometer with a spectral interval of 1 nm. Room temperature X-ray photoelectron spectra (XPS) were measured in an ultrahigh vacuum using a Thermofisher Escalab 250Xi X-ray photoelectron spectrometer. The excitation source used was monochromatized Al K$\alpha$ X-ray radiation. The crystal structures before and after irradiation with neutrons were determined with X-ray single crystal diffraction at room temperature using a Bruker APEX area-detector X-ray diffractometer equipped with Mo-K$\alpha$ radiation. The empirical absorption correction was employed with the SADABS program [12]. The structure was solved using direct methods and refined by full-matrix least-squares on F$^2$ with the SHELXTL97 software package [13,14].

## 3. Results and Discussion

*3.1. In-Situ and Real-Time Transmittance*

The in-situ and real-time transmittance of Nd:GSAG crystal varies with the fluctuation of neutron irradiation dose as shown in Figure 2a. In the first 65 min, with the increase in irradiation dose, the transmittance of the crystal decreased from 84 to 80%. Since the irradiation rate of the first 65 min is 7.3 mGy/s, the total irradiation dose is about 2.85 kGy. Interestingly, after 65 min, with the removal of the neutron irradiation, the transmittance of the crystal began to increase slowly with the increase in time. Finally, the transmittance of the crystal is stable at 83%, which is slightly lower than that of the crystal before irradiation. The results indicate that both transient and permanent optical loss were generated in the crystal after irradiation with the neutrons. This is in accordance with the results reported in the literature that transient and permanent optical loss were produced in optical materials after irradiation with gamma rays [1,15]. The possible mechanism for the transient optical loss is that the electrons, excited by the neutrons, are trapped during the generation of color centers and thus prevented from decaying back to lower stable states. This causes the crystal to become opaque and its transmittance reduced. At room temperature, after a fast thermal recovery, the transient color centers become unstable and the trapped electrons are able to decay back to the lower states [16]. Subsequently, the transient optical loss is dismissed. The simplified mechanism of the generation and dismissal of the transient optical loss are shown in Figure 2c,d.

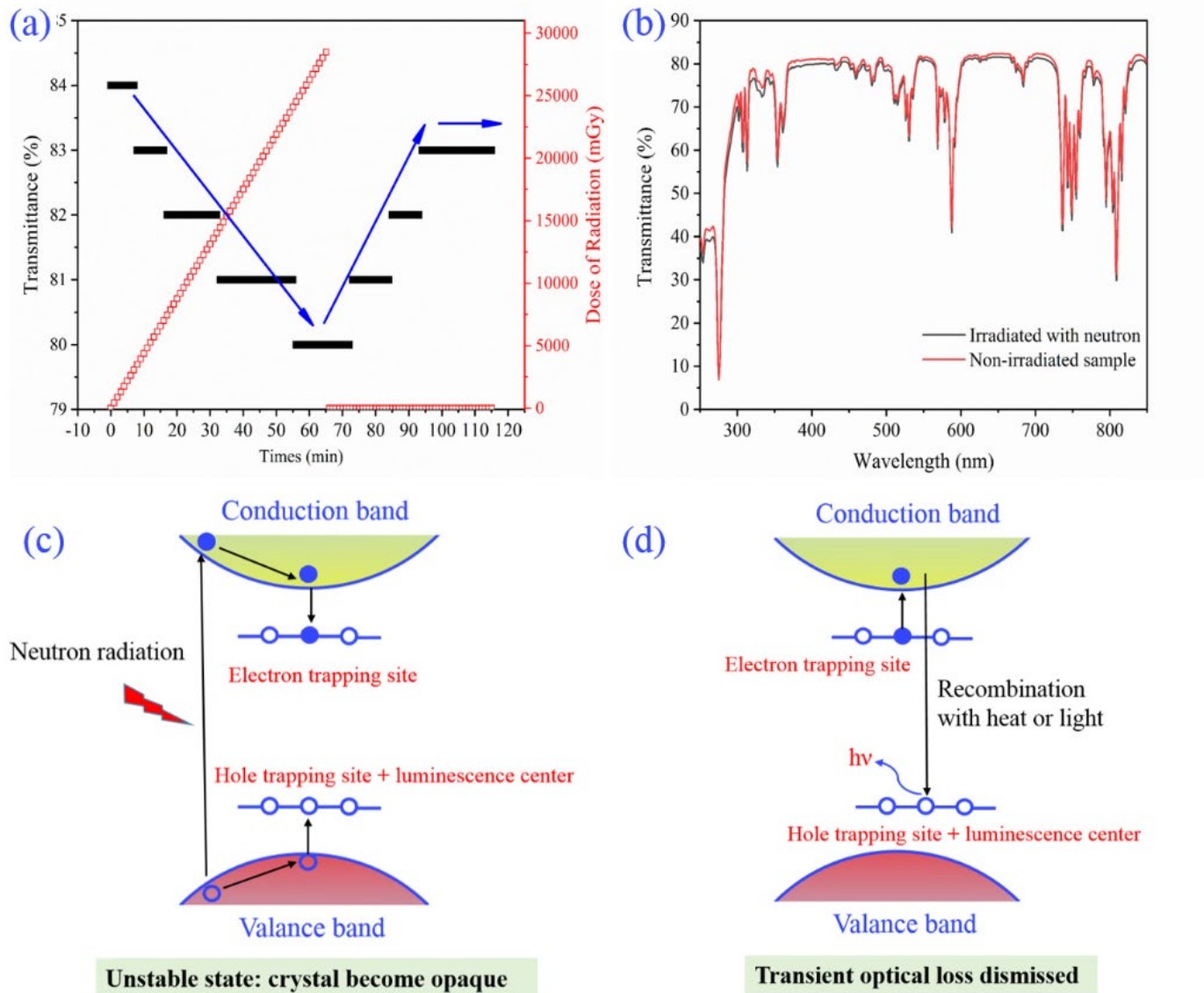

**Figure 2.** (**a**) The in-situ and real-time transmittance of Nd:GSAG crystal varies with the fluctuation of neutron irradiation dose. (**b**) The transmission spectrum of Nd:GSAG crystal before and after irradiation with 2.85 kGy doses of neutron. (**c**) Schematic representation of the trapping processes of electrons and holes with neutron radiation. (**d**) Schematic representation of the recombination processes of electrons and holes.

### 3.2. Optical Properties

The transmission spectra of Nd:GSAG crystal before and after irradiation with 2.85 kGy doses of the neutrons are shown in Figure 2b. The absorption peaks that appeared in the measured range have been assigned in our previous work [8,9]. As can be seen, the transmittance of Nd:GSAG crystal before and after irradiation almost remains constant in the measured range. It has been reported that the transmittance of Nd-doped phosphate glass decreases to half of the non-irradiated sample after irradiation with 5 kGy doses of gamma rays [6]. Thus, it can be concluded that Nd:GSAG crystal has excellent radiation resistance ability, which is consistent with the results of its irradiation with gamma rays that we reported previously [8].

### 3.3. Crystal Structure

The crystallographic data and details of the data collection of Nd:GSAG crystal before and after irradiation with neutrons are given in Table 1. It can be noted that the crystal structure remains constant (within experimental error) before and after irradiation. This result indicates that the crystal structure of Nd:GSAG after irradiation with neutrons was not damaged or changed. Thus, the transient and permanent transmittance reduction observed after irradiation with neutrons as shown in Figure 2 are more likely to be attributed to the formation of damage by ionization of electrons in the crystal [1].

**Table 1.** Crystallographic parameters of Nd:GSAG crystal before and after irradiation with neutrons.

| Formula | Non-Irradiated Sample | Irradiated with Neutrons |
|---|---|---|
| System and space group | Cubic, Ia-3d | Cubic, Ia-3d |
| a [Å] | 12.4265 | 12.4247 |
| V [Å$^3$] | 1918.87 | 1918.04 |
| F (000) | 1696 | 1696 |
| Reflections: measured/unique | 5263/187 | 2291/185 |
| R (int) | 0.0893 | 0.1018 |
| R (sigma) | 0.0308 | 0.0542 |

### 3.4. XPS Spectra

In order to study the valence state of the elements in Nd:GSAG crystal before and after irradiation with neutrons, high-resolution X-ray photoelectron spectroscopy (XPS) was performed. The XPS survey spectra of Nd:GSAG crystal before and after irradiation with neutrons are shown in Figure 3a. Concerning elemental composition, only C, O, Al, Sc, and Gd signals are detected on the sample surface. Auger transitions are also marked in the figure (centered at around 980 eV), but are not discussed, because it would be beyond the scope of this paper. More detailed narrow scan spectra of Gd 4d, Sc 2p, Al 2p, and O 1s core levels are presented in Figure 3b–e. Overall, the electron binding energies (BE) for Gd, Sc, and Al atoms are almost remaining constant before and after irradiation with neutrons. However, it is found that the peaks of O 1s core level shift to high BE with a shift amount of about 0.25 eV after the sample irradiation with neutrons. As we know, the XPS peak shift is a characteristic shift of the Fermi level [17]. It has been reported in the literature that the oxygen vacancy in oxide materials can push the Fermi level upward [18] and then increase the BE of O 1s core level. Thus, in this work, according to the characterization results of XPS, we can infer that permanent oxygen vacancies were produced in the crystal after irradiation with neutrons. Clearly, due to the crystal structure remaining constant before and after irradiation, the permanent reduction of the transmittance for the crystal after irradiation with neutrons can be attributed to the generation of permanent oxygen vacancy in the crystal.

In the case of a non-irradiated sample, a complex multiplet structure is observed for Gd 4d core level peaks which arise from electrostatic interactions between the 4d hole and 4f electrons. In particular, the BE signal is displayed at the characteristic shape of a spin-orbit doublet. Due to the spin-orbit interaction, the Gd 4d state is split into two Gd $^4$d$_{5/2}$ and $^4$d$_{3/2}$ peaks with a splitting energy of 4.7 eV, which is in agreement with that reported in the Gd$_2$O$_3$ host [19]. Further, an additional peak centered at 154.1 eV is observed because Gd belongs to the heavy lanthanide group element, so this splitting can be attributed to the electrostatic interaction between a core level and a partially filled 4f level [20]. The small shift of the main peaks is attributed to the relaxation effect [20]. The O 1s core level peaks are characterized by two main components. The strong peak centered at BE = 531.4 eV can be assigned to the presence of -OH groups and carbonates/bicarbonates, due to the very high reactivity of lanthanides towards $CO_2$ and $H_2O$ arising from atmospheric exposure [19,21]. The weak component centered at BE = 529.6 eV is related to oxygen in the Nd:GSAG crystal, which is in good agreement with the oxygen in other rare-earth oxides [22,23]. The Sc 2p state is also dominated by the two Sc $^2$p$_{3/2}$ and Sc $^2$p$_{1/2}$ components with a splitting energy

of 4.6 eV due to the spin-orbital interactions [24,25]. The strong component at BE = 401.5 eV is attributed to the Sc $^2$p$_{3/2}$, which is in good agreement with the value in pure Sc$_2$O$_3$ [24]. As shown in Figure 3e, only one component is observed for the Al 2p state, indicating that aluminum is completely oxidized in the compound [23]. This result is in complete accordance with that of Al in other Al-oxide compounds [26–28].

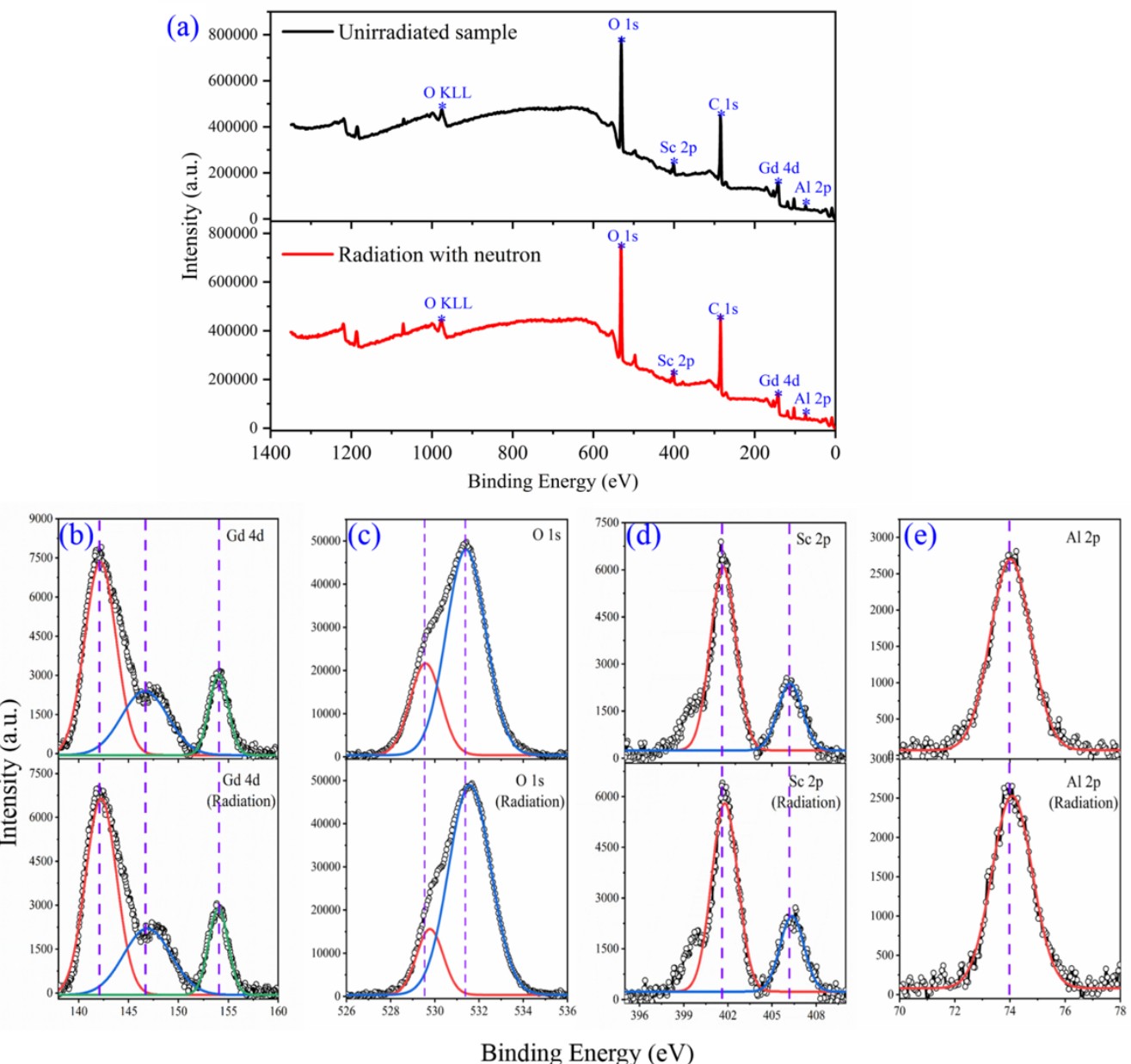

**Figure 3.** (**a**) The room temperature typical X-ray photoemission spectra (XPS) of Nd:GSAG crystal before and after irradiation with neutrons. (**a**) XPS survey spectra; (**b**) XPS spectra of Gd 4d core levels; (**c**) XPS spectra of O 1s core levels; (**d**) XPS spectra of Sc 2p core levels; (**e**) XPS spectra of Al 2p core levels.

## 4. Conclusions

In summary, the neutron irradiation resistance ability of Nd:GSAG single crystal has been investigated in real-time and in-situ using the High Intensity D-T Fusion Neutron Generator system. Both transient and permanent optical losses were observed in the crystal after irradiation with neutrons. X-ray single crystal diffraction results revealed that the crystal structure remains constant before and after irradiation with neutrons. It was found

by XPS that the peaks of O 1s core level shift to high binding energy with a shift amount of about 0.25 eV after the crystal irradiation with neutrons. Thus, all the results suggest that the reduction in the transmittance of the crystal after irradiation with neutrons can be attributed to the generation of oxygen vacancies in the crystal.

**Author Contributions:** Conceptualization, Y.G. and S.D.; Investigation, Y.G. and S.D.; Formal analysis, Y.G. and Y.C.; Writing—review and editing, Y.G. and S.D.; Supervision, Q.Z.; Project administration, W.L. All authors have read and agreed to the published version of the manuscript.

**Funding:** This research was funded by Plan for Anhui Major Provincial Science and Technology Project grant number (202203a05020002) and the National Natural Science Foundation of China grant number (52272011).

**Data Availability Statement:** Not applicable.

**Conflicts of Interest:** The authors declare that they have no known competing financial interests or personal relationships that could have appeared to influence the work reported in this paper.

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
