# Peer review of "Real-Time In-Situ Investigation of the Neutron Irradiation Resistance Ability of Nd3+-Doped Gd3Sc2Al3O12 Laser Crystal"

_crystals, doi:10.3390/cryst13010136_

Round 1

Reviewer 1 Report

The authors have reported on an investigation of the neutron irradiation resistance ability of Nd:GSAG crystal. The authors performed the absorption spectrum and the X-ray single crystal diffraction of the crystal structure before and after irradiation, and X-ray photoelectron spectroscopy of Nd:GSAG was also investigated with neutron radiation.

The results of this work will benefit the development of laser crystals. I recommend this work be accepted by Crystals after some revisions.

Minor issues:

1. Have the authors considered the influence of other defects produced during neutron irradiation on crystal properties?

2. Whether the dose of neutron irradiation used in the experiment meets the actual application requirements of the crystal?

3. The grammar in the article should be further improved, such as singular and plural.

4. Some references can be appropriately added to the introduction part of the manuscript.

Reviewer 2 Report

The manuscript is devoted to investigation of crystal structure, optical and XRD spectra of neutron irradiated Nd:GSAG crystal. Crystals of garnet family are widely used in lasers of wide spectral range. Knowledge about color center formation in GSAG crystal host affecting optical break down and laser operation efficiency may improve the functional characteristics of laser devices. Thus, the goal of investigation is actual. At the same time, there are some imperfections.

 Title. Full chemical formula of compound should be used in the manuscript title instead of acronym. Moreover, acronym should be used only after full chemical formula of compound in the text.

 Introduction. The authors claim that three types of trapping centers are formed under gamma irradiation. From my view point, these centers and their parameters have been briefly described.

 Experimental section. The pre-history of sample preparation affect dramatically color center formation. Can the authors tell anything about Nd:GSAG crystal preparation (crystal growth method, Nd concentration, purity of crystal)?

 Result and Discussion. Is there correlation between changes of optical and XRD spectra after gamma and neutron irradiation according to the previous investigation of the authors? Comparison of these data should be presented and discussed.

Does neodymium affect defect formation under neutron irradiation taking into account no small neutron cross section capture of its isotopes?

Did the authors use fresh cleavage or polished surface of the crystal for XRD measurements?

Round 2

Reviewer 2 Report

The comments “The authors claim that three types of trapping centers are formed under gamma irradiation. From my view point, these centers and their parameters have been briefly described” is without response
